# Tin(II) and Tin(IV) Complexes Incorporating the Oxygen Tripodal Ligands [(*η*^5^-C_5_R_5_)Co{P(OEt)_2_O}_3_]^−^, (R = H, Me; Et = -C_2_H_5_) as Potent Inflammatory Mediator Inhibitors: Cytotoxic Properties and Biological Activities against the Platelet-Activating Factor (PAF) and Thrombin

**DOI:** 10.3390/molecules28041859

**Published:** 2023-02-16

**Authors:** Alexandros Kalampalidis, Artemis Damati, Demetrios Matthopoulos, Alexandros B. Tsoupras, Constantinos A. Demopoulos, Gregor Schnakenburg, Athanassios I. Philippopoulos

**Affiliations:** 1Laboratory of Inorganic Chemistry, Department of Chemistry, National and Kapodistrian University of Athens, Panepistimiopolis Zografou, 15771 Athens, Greece; 2Department of Environmental Engineering, School of Engineering, University of Patras, 26504 Patras, Greece; 3Department of Forestry and Environment Natural Management, School of Plant Sciences, Agricultural University of Athens, 11855 Athens, Greece; 4Department of Biological Sciences, University of Limerick, V94 T9PX Limerick, Ireland; 5Health Research Institute, University of Limerick, V94 T9PX Limerick, Ireland; 6Bernal Institute, University of Limerick, V94 T9PX Limerick, Ireland; 7Department of Chemistry, International Hellenic University, 57001 Kavala, Greece; 8Laboratory of Biochemistry, Department of Chemistry, National and Kapodistrian University of Athens, Panepistimiopolis Zografou, 15771 Athens, Greece; 9Institut für Anorganische Chemie, Rheinische Friedrich-Wilhelms-Universität Bonn, Gerhard-Domagk-Straße 1, D-53121 Bonn, Germany

**Keywords:** oxygen tripodal ligand, Kläui ligands, Sn(II) and Sn(IV) compounds, antiplatelet drugs, cytotoxicity, platelet-activating factor (PAF), thrombin

## Abstract

Metal complexes displaying antiplatelet properties is a promising research area. In our methodology, Platelet-Activating Factor (PAF), the most potent lipid pro-inflammatory mediator, serves as a biological probe. The antiplatelet activity is exerted by the inhibition of the PAF-induced aggregation in washed rabbit platelets (WRPs) and in rabbit plasma rich in platelets (rPRPs). Herein, the synthesis and biological investigation of a series of organometallic tin(II) and tin(IV) complexes, featuring the oxygen tripodal Kläui ligands [(*η*^5^-C_5_R_5_)Co{P(OEt)_2_O}_3_]^−^, {R = H, (L_OEt_^−^); Me (L*_OEt_^−^)}, are reported. Reaction of NaL_OEt_ (**1a**) and NaL*_OEt_ (**1b**) with SnCl_2_, yielded the rare four-coordinate L_OEt_SnCl (**2a**) and L*_OEt_SnCl (**2b**) complexes. Accordingly, L_OEt_SnPh_3_ (**3a**) and L*_OEt_SnPh_3_ (**3b**) were prepared, starting from Ph_3_SnCl. Characterization includes spectroscopy and X-ray diffraction studies for **2a**, **2b** and **3b**. The antiplatelet activity of the lead complexes **2b** and **3a** (IC_50_ = 0.5 μΜ) is superior compared to that of **1a** and **1b**, while both complexes display a pronounced inhibitory activity against thrombin (IC_50_ = 1.8 μM and 0.6 μM). The in vitro cytotoxic activities of **3a** and **2b** on human Jurkat T lymphoblastic tumor cell line is higher than that of cisplatin.

## 1. Introduction

Undoubtably, over the last decades the coordination and organometallic chemistry of tin has been extensively studied [1,2]. A plethora of organotin(IV) complexes is known, which display diverse structural characteristics along with several applications [3]. For example, these extend from catalysis agents [4], biocides [5] and antimicrobials [6], to antifouling agents or paints [7]. In particular, organotin(IV) complexes have been potentially used as anticancer agents in cancer chemotherapy, due to their ability to interact with DNA [8,9]. They display a different profile compared to standard platinum cytotoxic complexes, while one of the main drawbacks for further biological applications remains the acute toxicity of organotin compounds [10]. An excellent review article by Banti et al. [11] highlights the recent progress in the area, emphasizing the anti-proliferative activity of organotin(IV) complexes.

On the other hand, careful synthetic design based on the incorporation of bulky substituents around the ligand periphery has led to stabilized divalent tin(II) monomeric and dimeric species [12,13,14].

The continuous and systematic interest towards this class of complexes is further reinforced by the structural diversity and reactivity patterns of the related species. The sterically demanding Kläui ligand [(*η*^5^-C_5_R_5_)Co{P(O)(OEt)_2_}_3_]^−^, {R = H, (L_OEt_^−^) [15], Me (L*_OEt_^−^) [16]} (Figure 1a) is suitable for the selective formation of an appropriate four-coordinate divalent species of the group 14 heavier analogs [17].

It belongs to a class of monoanionic, tridentate oxygen ligands whose coordination chemistry has been studied in depth, including a wide variety of main groups, transition metal ions in low and high oxidation states as well as f-block elements [18,19]. The chemical inertness of this ligand together with its thermal and hydrolytic stability is of crucial importance, especially in the field of organometallic chemistry [20]. Furthermore, it has been used as an ionophore for the efficient transportation of alkali metal ions (Li^+^) through phospholipid membranes [21].

To this end, our research emphasizes the biological role of transition metal complexes and their therapeutic potentials as antiplatelet and/or antithrombotic agents. Our approach focuses on the Platelet-Activating Factor (PAF), that serves as a biological probe (Figure 1b). It is a phospholipid signaling molecule of the immune system and a potent mediator of inflammation [22,23,24]. The biological activity of PAF is exerted upon binding to its G-protein coupled receptor PAFR (PAF-receptor), which is found on the plasma membrane of cells or intracellular membranes [24]. In addition, PAFR is considered a lead drug target for treatment of inflammation and asthma along with cardiovascular diseases [24].

Following this strategy, the antiplatelet (anti-PAF) behavior of a new compound can be evaluated using the most internationally accepted methodology for studying the in vitro inhibition of PAF activity. That is the potency to inhibit the action of the PAF-induced aggregation in washed rabbit platelets (WRPs). As a result, we have presented a series of metal complexes with potent antiplatelet activities that inhibit the PAF-induced aggregation in the nanomolar and sub-micromolar level [25,26,27]. Consequently, preliminary structure–activity relationships have been established that are summarized in a review article, a patent, and a book chapter contribution [28,29,30]. Notably, the antiplatelet activity of certain metal-based PAF-induced inhibitors was comparable or even higher than that of known organic small molecules (WEB 2170, BN 52021, CV-3988, etc.) with potencies in the sub-micromolar range [29,30,31,32]. In the meantime, this topic has collected interest from other research groups [33,34,35]. Interestingly, the term “inflammatory mediator inhibitors” has been coined in a recent review article, highlighting our contribution in the field [36].

Herein, we report on the synthesis of two four-coordinate tin(II) complexes (**2a**, **2b**) and two six-coordinate organonotin(IV) complexes (**3a**, **3b**), sterically encumbered by the spectator oxygen tripodal ligands NaL_OEt_ (**1a**) and NaL*_OEt_ (**1b**). Emphasis is given to the study of the antiplatelet and antithrombotic activity of NaL_OEt_ and NaL*_OEt_ ligands, the tin(II) and tin(IV) precursors and their combination to afford the target complexes **2**–**3**. All complexes have been tested as inhibitors against the PAF-induced aggregation in washed rabbit platelets (WRPs) and in rabbit PRPs (rPRPs) and thrombin-induced WRPs aggregation. To our knowledge, this is the first time that the antiplatelet and antithrombotic activities of the compounds reported herein were studied. The in vitro cell growth inhibition of the new organotin(II), and tin(IV) complexes, and the ligand and metal precursors, have also been investigated against Jurkat T lymphoblastic tumor cell line.

## 2. Results and Discussion

### 2.1. Chemistry

Mononuclear Sn(II) and Sn(IV) complexes, incorporating the Kläui oxygen tripodal ligand, have been prepared in high yields starting from SnCl_2_ and Ph_3_SnCl, respectively (Figure 1).

The reaction of SnCl_2_ with one equivalent of NaL_OEt_ (**1a**) or NaL*_OEt_ (**1b**) in THF at low temperature (−40 °C to −60 °C), resulted selectively in the formation of the tin(II) chlorides **2a** and **2b** as yellow microcrystalline solids and in yields ranging from 93–95%. Both solids are air-sensitive and dissolve readily in CH_2_Cl_2_, CHCl_3_ and THF and moderately in diethyl ether and pentane. In CHCl_3_, **2a** and **2b** are stable at ambient temperature, under an argon atmosphere, for at least two days as proven by ^1^H, ^31^P NMR spectroscopy. They melt without decomposition at 156 °C and 199 °C, respectively. The identity of **2a** and **2b** was established by a combination of FT-IR (Appendix A), multinuclear NMR spectroscopy, satisfactory elemental analyses and single-crystal X-ray crystallography.

The FT-IR spectrum of **2a** and **2b** display characteristic ν(C-H) aliphatic stretching vibrations at 2977 cm^−1^ and 2970 cm^−1^, in accord with the literature [37]. For **2b,** the ν(C-H) aromatic stretching vibration appears at 3118 cm^−1^. Additionally, the bands in the region of 1600–1350 cm^−1^ are typical for the ring stretching vibration modes ν(C=C), of the cyclopentadienyl ligand and the ν(C-C) mode of the -P(O)(OEt)_2_ moiety.

The FT-IR spectra in KBr show two strong absorptions for the *δ*(P=O) deformation vibrations at 591 and 576 cm^−1^ (**2a**), and 595 and 578 cm^−1^ (**2b**). A *C*_3v_-symmetric coordination of both ligands to the tin(II) center is adopted [38], which is verified from the crystal structures of **2a** and **2b** showing one long and two considerably shorter Sn–O bonds. In the region of 1160–938 cm^−1^, several very strong *ν*(P=O) absorptions are observed and full assignment is rather difficult. However, a comparison of these absorptions, with those of **1a** (1171 cm^−1^) and **1b** (1158 cm^−1^), allows the assignment of one of the expected *ν*(P=O) absorptions at 1119 cm^−1^ and 1106 cm^−1^ to complex **2a** and **2b**, respectively. For both complexes, a profound shift (Δν = 50 cm^−1^, compared to that of **1a** and **1b**) of the *ν*(P=O) absorption towards the lower wavenumber’s region, is observed, confirming coordination to the tin(II) center via the oxygen atom of the P=O group. For **2a** the absorption at 844 cm^−1^ could be assigned to the γCH vibration mode of the cyclopentadienyl ligand [39].

The molecular structures of **2a** and **2b** as determined by single-crystal X-ray diffraction are depicted in Figure 2 and Figure 3. Suitable single crystals were obtained upon slow cooling of a saturated diethyl ether solution of **2a** or **2b** at −30 °C. Complex **2a** crystallizes in the triclinic crystal system and space group P−1, while **2b** in the monoclinic space group P2_1_. Selected bond lengths and angles, with estimated standard deviations for **2a** and **2b**, are included in Appendix A.

The asymmetric unit of compound **2a** contains two crystallographically independent molecules with marginally different bonding parameters. In the asymmetric unit, molecules of **2a** are interconnected via non-classical hydrogen bonding interactions, as depicted in Appendix A. These, for example, include typical intramolecular (C10-H10B∙∙∙Cl1 = 3.140 Å; bond angle = 172.81°; C14-H14A∙∙∙O7 = 2.922 Å; bond angle = 105.04°) and intermolecular hydrogen bonding interaction (C16-H16B∙∙∙Cl2 = 2.844 Å; bond angle = 144.46°; C17-H17Β∙∙∙Cl2 = 3.499 Å; bond angle = 105.50°), respectively. For clarity, only one molecule has been drawn in Figure 2. The thermal ellipsoids of the carbon atoms of the C_5_R_5_ ring and of the phosphonato groups {P=O(OEt)_2_} in both molecules are disordered. This feature is quite common in related complexes, such as [(L_OEt_)_2_Cu] [40] and [(L_OMe_InTp)][InCl_4_] [41].

Complexes **2a** and **2b** adopt a “see-saw” shape, which is the geometry derived from the trigonal bipyramid. The two axial positions are occupied by a chlorine atom and one oxygen atom of the relevant ligand L_OEt_^−^ (L*_OEt_^−^), while the tin lone pair and the other two oxygen atoms of the ligand are in the equatorial positions. For **2a,** distortion from the ideal trigonal bipyramidal geometry is evident by the relevant O(1)-Sn-O(3), O(2)-Sn-O(3) and O(2)-Sn-O(1) bond angles of 76.63(8)°, 78.49(7)° and 89.93(8)°, respectively. This is also shown from the O(3)-Sn-Cl bond angle of 157.85(5)°, which deviates from the ideal of 180°. The Sn-Cl bond distance in **2a** (258.47(10) pm) and **2b** (262.4(2) pm) is shorter than those reported for other tin(II) chlorides (e.g., CpSnCl [Sn-Cl = 267.9(5) pm] [42], Cp^*^SnCl [average Sn-Cl = 267.5(1) pm]) [43]. It compares well with the bond length of 262.9(1) pm reported for the TpSnCl analog (Tp stands for tris-pyrazolyl borate) [44]. The structural features of complexes **2a** and **2b** are in good agreement with those reported for the relevant germanium (II) analog L_OEt_GeCl [17]. For **2b**, stabilization in the unit cell is provided by the intermolecular non-classical hydrogen bonding interactions, C15-H15∙∙∙Cl2 = 2.992 Å; bond angle = 163.02° and C21-H21A∙∙∙O9 = 2.479 Å; bond angle = 163.02°, respectively (Appendix A).

The sterically demanding Kläui-type ligands L_OEt_^−^ (L*_OEt_^−^) have also been employed for the preparation of the corresponding Sn(IV) complexes. Consequently, the reaction of Ph_3_SnCl with **1a** (**1b**) in THF yielded straightforwardly the complexes **3a** (**3b**) at 75% yield, after crystallization from CH_2_Cl_2_/*n*-C_5_H_12_ at low temperature (Figure 1). Both complexes are air stable, in the solid state, and melt without decomposition at 232 °C and 278 °C, respectively. In CHCl_3_ and at ambient temperature, under an argon atmosphere, they are stable for at least two days, according to NMR spectroscopy. Both complexes were fully characterized by means of FT-IR (Appendix A) and multinuclear NMR spectroscopies (Appendix A). The known complex **3a** has been prepared for comparison [45]. The synthetic procedure reported herein was performed in THF (CH_2_Cl_2_ in the published procedure), affording **3a** in 75% yield. The spectroscopic data of **3a** compare well with those of the published procedure.

The FT-IR spectra of **3a** and **3b** display characteristic medium intensity bands for the ν(C-H) aromatic stretching vibration, at approximately 3060 cm^–1^, while the ν(C-H) aliphatic streching vibrations are present in the expected region. At this point, it must be noted that the IR spectra of both complexes are dominated by very strong absorption bands, in the region of 1250 cm^–1^ to 500 cm^–1^. The FT-IR spectra in KBr also show two strong absorptions for the *δ*(PO) deformation vibrations at 615, 581 cm^−1^ (**3a**) and 622, 587 cm^−1^ (**3b**), while the *ν*(PO) absorptions appear at 1121 cm^−1^ (**3a**) and 1112 cm^−1^ (**3b**). These wavenumbers are considerably shifted at lower wavenumbers, in comparison to that of **1a** (Δν = 50 cm^−1^) and **1b** (Δν *=* 46 cm^−1^), suggesting coordination of the (P=O) group, to the tin(IV) center. These *ν*(PO) absorptions appear at significantly lower wavenumbers compared to that reported for {[C_6_H_2_[P(O)(OEt)_2_]_2_-2,6-*t*-Bu-4-Ph_3_Sn} [46] at 1233 cm^−1^, indicating stronger P–O bonding interaction in the latter.

Suitable single crystals of **3b** were obtained upon slow cooling of a saturated toluene solution of **3b** at 4 °C over 1–2 h, and then at −40 °C for a period of 2 days. Complex **3b** crystallizes in the triclinic crystal system and space group *C*2/*c*. The molecular structure of **3b** is depicted in Figure 4, while selected bond lengths and angles are included in Appendix A. Complex **3b** exhibits a distorted octahedral coordination geometry, as is verified by the characteristic O(1)-Sn-C(29), O(2)-Sn-C(23) and O(3)-Sn-C(35) bond angles of 86.2(2)°, 87.7(2)° and 159.7(2)°, respectively. The structural features of **3b** are in accord to those reported for complex **3a** [45]. In addition, in the unit cell pairs of **3b** arranged in a head-to-tail fashion are stabilized by characteristic intermolecular C-H∙∙∙π interactions including the hydrogen atom H(22C) of the -PO(C_2_H_5_)_2_ group from one molecule, and the centroid of an adjacent molecule ((C22-H22C)∙∙∙centroid (C29-C34)), at a distance of 2.936 Å. Further stabilization is due to the intramolecular C-H∙∙∙π interaction between H(11A) and the ring centroid (C29-C34) (distance of (C11-H11A)∙∙∙centroid (C29-C34) is 2.926 Å) (Appendix A).

### 2.2. Evaluation of Biological Activity

The biological assays (anti-PAF and cytotoxicity studies) of the four-coordinate tin(II) complexes **2a** and **2b** were cautiously carried out in dimethyl sulfoxide (DMSO) since they do not dissolve in water. Stability in this solvent was checked by means of ^1^H and ^31^P NMR spectroscopy, revealing that **2b** is more stable than **2a** (Appendix A). For **2a**, and upon standing in DMSO-d_6_ for some hours, new species were formed (Appendix A), that could be attributed to DMSO adducts [47]. The solubility of **3b** in this solvent is rather poor*,* while for **3a** the formation of new species became evident after approximately 12 h (Appendix A). To avoid this obstacle, the relevant complexes were dissolved in DMSO immediately prior to use.

#### 2.2.1. Inhibitory Effects against the Biological Activities of PAF in Washed Rabbit Platelets (WRPs) and in Rabbit PRPs (rPRPs)

In this study, we report on the inhibitory effect of the organotin(II) and organotin(IV) complexes bearing the oxygen tripodal Kläui ligands, against the PAF-induced aggregation in WRPs and rPRPs. The inhibitory effect of all the complexes tested, expressed as IC_50_ values in μM, is shown in Table 1.

Apparently a low IC_50_ value reflects strong inhibition of the PAF-induced aggregation for a given PAF concentration. For comparison, the inhibitory effect of the known q1anticancer reagent cisplatin is included (See Section 2.2.4). To our knowledge this is the first time that organotin-based complexes were tested as antiplatelet and antithrombotic agents. This is also the case for the relevant oxygen tridentate ligands 1a and 1b.

Based on the results presented in Table 1, varying degrees of inhibition against the PAF/PAF-R and thrombin/PAR-1 in either WRPs or rPRPs were observed.

##### Inhibitory Effects against the Biological Activities of PAF in Washed Rabbit Platelets (WRPs)

In WRPs, SnCl_2_ × 2H_2_O displayed a weak inhibitory effect (63 μM) in comparison to Ph_3_SnCl and **1a** and **1b**, which are active in the micromolar range (IC_50_∼1 μM). The results listed in Table 1 further indicate that within the complexes tested, the permethylated organotin(II) analog **2b** and the organotin(IV) complex **3a**, which incorporates the non-methylated L_OEt_ ligand, were the most potent against PAF-induced aggregation in WRPs. The new complexes **2b** and **3a** display an inhibitory effect in the micromolar range and in a dose-dependent manner. Their IC_50_ values were significantly lower than those of all the other complexes tested. In fact, ongoing from **1b** to **2b** and from **1a** to **3a**, the IC_50_ drops by half (from ∼0.9 to ∼0.5 μΜ), accompanied equally, in the case of **3a**, by a two-fold decrease of the relevant activity of Ph_3_SnCl (0.88 μΜ). Notably, and upon coordination of **1a** and **1b** to SnCl_2_ × 2H_2_O, a six-fold decrease of the IC_50_ value is observed for **2a** and a pronounced two-orders of magnitude drop for the case of **2b**, in comparison to the IC_50_ value of SnCl_2_ × 2H_2_O (63 μM). Since the final products **2b** and **3a** showed the lowest IC_50_ values against the PAF-induced aggregation in WRPs, it becomes evident that coordination of the bioactive oxygen tripodal ligand L_OEt_ (L_Oet_*) into the metal precursors SnCl_2_ × 2H_2_O and Ph_3_SnCl, respectively, induces a significant increase of the anti-PAF potency of these organometallic complexes. This is in accordance with the well-established general trend known as the *synergetic effect*, which refers to the positive influence of ligand coordination to a metal center on the antiplatelet potency of the investigated complex [29]. This trend has been verified upon examination of a series of metal-based analogs [29]. The effect of separate metal ions and the ligand molecules was always performed; thus, the observed effect is not a cumulative effect of ligand molecules and metal ions, but a combination of both substances affording the relevant metal complexes.

With respect to the anti-PAF (antiplatelet) potency of the organotin(II) complexes in WRPs, better outcomes were observed with **2b** that bears the L_OEt_* bioactive methylated form of the ligand, since it displays at least one order of magnitude lower IC_50_ value than its non-methylated analog **2a**. Interestingly, opposite outcomes were observed for the organotin(IV) complexes. Complex **3a** showed much stronger anti-PAF effects against PAF-induced aggregation in WRPs, as its IC_50_ value was one order of magnitude lower than that of **3b**. It seems that the co-presence of the three phenyl groups and the sterically encumbered L_OEt_* ligand around the tin(IV) center, renders **3b** more bulky and thus not suitable for selective binding to the PAF-receptor (PAFR) [25]. As a result, the anti-PAF bioactivity of **3b** is reduced. In the case of **2b**, however, the presence of the methylated L_OEt_^*^ ligand results in an increase of its anti-PAF bioactivity. This discrepancy may be attributed to structural differences of the ligand upon binding to PAFR. To this end, preliminary theoretical docking calculations performed corroborate with the previously reported results [29], based on the size of incoming substance (complexes **2b, 3a** and **3b**), towards the binding site of the PAFR (entry 18, Appendix A). Apparently, these results clearly denote that the biological effect exerted is due to the overall structural characteristics of the organotin complexes bearing the parent oxygen tripodal ligand. Conclusively, we may propose that within this series the inhibitory activity drops in the following order: **2a** > **3b** > **3a** ≅ **2b**.

Since there are no reports on similar studies with other organotin complexes, a direct comparison of the anti-PAF activities of our complexes cannot be made in a straightforward manner. In any case, the inhibitory effects (IC_50_ value) of the organotin(II) and (IV), are similar to that reported for several well-established Rh(III) (IC_50_ = 0.1–2.6 μM) [25], Ir(III) [8 μM] [48] as well as for Ru(II) and Ru(III) (IC_50_ = 0.2–7 μM) PAFR antagonists, with potencies in the micromolar range [49]. Notably, their biological activity against the PAF-induced aggregation in WRPs is comparable with the inhibitory effect of well-established natural PAF antagonists/inhibitors from the series of Gingolides B, namely BN 52,020 (IC_50_ = 3.6 μM), BN 52,021 (IC_50_ = 9.7 μM) and BN 52,022 (IC_50_ = 38 μM) [29]. Furthermore, the anti-PAF potency of the relevant organotin complexes could be compared to that of Rupatadine, with an IC_50_ value of 260 nM against the PAF-pathway.

##### Inhibitory Effect towards the PAF-Induced rPRPs Aggregation

Ιn rPRPs, as expected, the IC_50_ values were higher (lower anti-PAF activity) by at least an order of magnitude, in relation to the WRPs values. Notably, even in the presence of plasma **2b** remains the most potent, followed by **3b** and **3a**, with **2a** being the least active. It is also evident that in conditions close to the real ones, the anti-PAF bioactivity of **2b**, upon ligand coordination to the metal precursors, is enhanced (*synergetic effect*). This behaviour corroborates with the anti-PAF activities of other known Rh(III)- and Ru(II, III)-based inhibitors reported in the literature [25,27,29,49].

Cautiously, and ignoring the presence of plasma proteins and other biomolecules that potentially interact with the incoming substances, the observed IC_50_ values of the organotin(II) and organotin(IV) complexes against the PAF-induced aggregation in rPRPs, could be compared with those of known metal complexes, tested in human PRPs [27,34]. Interestingly, the potency of **2b** (IC_50_ = 4.5 μM) is in the same range with that of a series of gold(I) N-heterocycle carbene (NHC) complexes (IC_50_∼1–3 μM) [34], and significantly lower (almost two orders of magnitude) than that of the Rh(I) analog [Rh(cod)Cl(tpc)] (IC_50_ = 122.6 μM; tpc is a substituted thieno-pyrimidine ligand) [27]. In particular, the activity of **2b** is also higher compared to that of the [Ru(η^6^-cymene)(L)Cl] complex (L is chrysin, a natural flavonoid), exerting a non-concentration-dependent inhibition in the range of 100–500 μΜ [50]. Remarkably, complexes **2b**, **3a** and **3b** display significantly higher biological activity (inhibitory effect) compared to that of the well-known antiplatelet drug aspirin, with an IC_50_ value of 291 μΜ in hPRPs, and ginkgolide B (IC_50_ = 65.4 μΜ), a specific inhibitor of PAF-induced platelet aggregation in PRPs [51].

The above reported results strongly suggest that although in vitro studies in WRPs remain the more studied, studies in rPRPs, such as those performed herein, have their own importance for preclinical evaluation of the health benefits or side-effects of an antiplatelet compound on platelet function before its in vivo use and ex vivo clinical research. In fact, the parent organometallic complexes (**2b**, **3a** and **3b**), remain active in more physiological media, while the ligands and the metal precursors are mostly affected by plasma.

#### 2.2.2. Inhibitory Effect against Thrombin

Based on the interesting results reported for the new organotin complexes against the PAF-induced aggregation in WRPs, we further studied their antithrombotic potency. The four-coordinate organotin(II) complex **2b** showed a considerably stronger antithrombotic effect compared to the ligand **1b** and the metal precursor. However, ongoing from **1a** to **2a**, the antithrombotic potency decreases by a factor of ten. Remarkably, this trend is also followed in the case of WRPs (part 2.2.1.1) and could be possibly attributed to the presence of the non-methylated ligand precursor L_OEt_, in the organotin(II) complex **2a**. As for the PAF-induced aggregation, we realized that the relative activity trend is followed in the case of thrombin as well. Thus, **2b**, which incorporates the L_OEt_* moiety, was the most potent (IC_50_ = 1.8 μΜ).

The six-coordinate organotin(IV) analogs **3a** and **3b** seem to respond differently against the thrombin/PAR-pathways. Both organometallic complexes display the strongest antithrombotic effect with IC_50_ values in the sub-micromolar range (0.55 μΜ for **3a** and 0.23 μΜ for **3b**). These values were almost 20- to 10-fold lower than those of **2a** and **2b**. Interestingly, **3b** presents a profound 70-fold decrease in its activity when compared to **1b** and about an order of magnitude lower activity, than that of the metal precursor Ph_3_SnCl. Presumably these organotin complexes exert a higher affinity for the PAR receptors of thrombin, rendering them strong inhibitors of the thrombin pathways. This is of interest, since previous reports with the [Ru(η^6^-cymene)(L)Cl][BF_4_] complex (L stands for 4-phenyl-2-pyridin-2yl-quinazoline) in washed human platelets, inhibit platelet aggregation at a maximum concentration of 500 μΜ [52].

#### 2.2.3. Desensitization Tests

Interestingly, the results of the present study revealed that the ligand precursor **1b** and the organotin(II) analog **2b** induced platelet aggregation in WRPs in higher concentrations. In the cross-desensitization tests [53,54], platelets aggregated by these compounds were not reactivated by PAF, and inversely, administration of these compounds could not reactivate platelets that were firstly aggregated by PAF (Figure 5). For this reason, the PAF-pathway-related aggregatory effects of the samples tested on rabbit platelets are expressed as the concentrations of PAF that can induce the same effect of 50% of platelet aggregation. This is known as the concentration effective in producing 50% of the maximum response (EC_50_ values). More specifically, the EC_50_ values of **1b** and **2b** were 131 μM and 75.8 μΜ, respectively. In comparison to their highest IC_50_ values against platelet aggregation, higher amounts of **1b** and **2b** (at least one to two orders of magnitude) were required to induce reversible agonistic aggregatory effect in WRPs aggregation, with a similar potency to that of the PAF-induced platelet aggregation. However, in the concentration range of approximately 250–500 μM, both substances caused an irreversible platelet aggregation and formation of toxic permanent clots. Therefore, we may notice that at high concentration levels, both compounds, could have a rather weak agonistic activity through the PAF/PAF-R pathway, but at lower levels, their inhibitory activities against PAF prevail.

The reported results from the cross-desensitization tests further suggest that **2b** induces aggregation of WRPs through the PAF/PAFR-related pathway of platelet aggregation. Obviously, **1b** and **2b** are considered potent PAF-inhibitors since either antagonistic (in low amounts) or agonistic (in higher concentrations) behaviour can beneficially affect the PAF/PAFR-induced inflammation and thrombosis. This occurs by disallowing the far more active PAF-molecule to act through its PAFR-related inflammatory pathways.

#### 2.2.4. Cell Viability Assay

Following the genotoxic and cytotoxic effects previously reported for the organotin complexes **2**–**3**, and their precursors NaL_OEt_ (**1a**), NaL*_OEt_ (**1b**), SnCl_2_ × 2H_2_O and Ph_3_SnCl on peripheral healthy blood lymphocytes [55], in this report, cell viability was determined by means of Trypan blue exclusion assay, against the Jurkat T lymphoblastic tumor cell line [56]. The test was performed twice with particular attention on the protocol used. Cell suspensions were microscopically observed, and the percentage of dead cells was estimated. Data are presented as histograms in Figure 6 and in Appendix A, summarizing the viability rates of Jurkat cells’ culture in the presence of the tested organotin compounds, their precursors and cisplatin that served as a positive control.

Jurkat cell cultures were given 17 h to start growing in the multi-well plates before the various organotin complexes were added in the culture medium, immediately after dissolution in DMSO. The number of cells per well, before the addition of the appropriate compound, was considered as the 100%. The various complexes were left to interact with the cells for 3 h before the recovery period. During the recovery period, cells that may have suffered membrane damage but were not killed have the time to undergo membrane reparation, while the recovery period was enough for the cells to undergo a complete cell cycle. At 48 h post-culture initiation, the number of cells was measured in order to evaluate the viability percentage. During the experimental procedure, the number of cells per mL was measured at 17, 20 and 48 h post-culture initiation. The comparative analysis was based on the number of cells at 17 h post-culture initiation and the number of cells at the end of the culture period [55].

A fluctuation in the viability of Jurkat cells in the absence of any tested compound is evident, taking into consideration the 0 μM concentration. This is normal as it is not possible to seed the same number of cells per well. Cisplatin (control) was studied at the same concentrations as the organotin complexes and their precursors, so that the results are comparable. Considering the well-reported anti-proliferative activity of cisplatin, at the concentration of 75 μM the percentage of viable cells reached to 4.8% after three hours exposure in a concentration-dependent way.

The effect of SnCl_2_ × 2H_2_O is also concentration dependent (Figure 6 and Appendix A) and varies from 153.8% (1 μM) to 27.2% (75 μM), with the higher concentration (75μM) showing activity comparable to that of cisplatin. On the other hand, Ph_3_SnCl remains lethal (5.80%) towards Jurkat cells, even from the lowest concentration (1 μM), which is indicative of its high toxicity in this cell line. Similar effects were observed on peripheral blood lymphocytes [55].

Notably, the amphiphilic spectator ligand precursor **1a** (NaL_OEt_) appears not to affect the viability of the present cell line, while for the permethylated analog **1b** (NaL*_Oet_), loss of viability was not dose dependent. It seems, therefore, that the presence of methyl groups in the cyclopentadienyl ring reduces to some extent, the Jurkat’s cell viability. In any case, the viability percentage was higher than that of cisplatin.

Among the organotin complexes carrying Cl ligands, **2a** shows good activity in the range of 1–20 μM, slightly higher compared to the control (exception at 5 μM), while **2b** displays a significant reduction of the cell viability upon increase of the concentration. Cell line viability (%) when using **2b** (at concentrations > 5 μM) is lower than for cisplatin. In any case, **2b** appeared to be more cytotoxic than **2a**, in the concentration range of 10 to 75 μM. On peripheral blood lymphocytes, the cytotoxic activity of **2a** begins at the concentration of 20 μM with statistical gravity (*p* < 0.001) at concentrations of 50 and 75 μM. This observation is compatible with earlier reposted observations [55].

The cell growth inhibitory profile of the triphenyltin(IV) complexes **3a** and **3b** differs significantly. From the results depicted in Figure 6, it is clearly seen that **3a** is the most active, displaying similar cell growth inhibition to that of the Ph_3_SnCl precursor in the concentration range of 20 μM–75 μM. It is to be mentioned that inhibition occurs, even at the lower concentrations (1 and 5 μM). On the other hand, the methylated analog **3b** affects cell viability in a concentration-dependent manner. To this end, it seems that the presence of methyl groups in the cyclopentadienyl ring improves the cell viability percentage (%) of this complex, in comparison to **3a**, within the whole concentration range. Interestingly, **3a** displays viability rates lower than the ones of the control, within the concentration range 1–75 μM, which is not the case for **3b**. This indicates that **3a** is very potent against this cell line. Compared to the lymphocyte cultures, **3a** at the lowest concentration (1 μM) affected Jurkat cells’ viability, while **3b** caused cytotoxicity in peripheral blood lymphocytes already at the concentration of 5 μM [55,57].

The results reported herein are in good agreement with previously reported studies, mainly concerning organotin(IV) complexes [58,59,60].

Overall, the cell viability results performed further reveal that **2b**, a potent anti-PAF inhibitor (IC_50_ = 0.5 μM), is more cytotoxic compared to cisplatin, which displays an IC_50_ value of 0.55 μM against PAF. Thus, the viability percentage drops dramatically (from the concentration of 10 μM), compared to that reported for the control (Figure 6). On the other hand, **3a** a sub-micromolar PAF-induced aggregation inhibitor in WRPs (IC_50_ = 0.5 μM), was proven to be very potent against the present cell line, even from the concentration of 1 μM.

In addition, recent reports in the field demonstrate that in severe inflammatory procedures implicated in cancer situations such as melanoma, the PAF- and thrombin-activated pathways are interrelated [61]. Moreover, it has been suggested that PAFR antagonists (WEB2086), administered in combination with chemotherapy, may represent a promising strategy for cancer treatment [62]. In other words, in vivo administration of PAF receptor antagonists may reduce the formation of new vessels in cancer cells [63].

To this end it became apparent that the lead complexes **2b** and **3a** could be considered as possible antiplatelet, antithrombotic and anticancer agents. In both substances, the increased cytotoxicity observed in Jurkat cells corroborates with their potent anti-PAF (anti-inflammatory) activity. This is of interest since administration of a substance with dual anti-PAF and cytotoxic activity could inhibit tumor development, decreasing cell proliferation, as already reported in vivo, after animal treatment with a combination of cisplatin and a PAF inhibitor [62].

In any case, additional experiments are required to gain insight on the pharmacological profile of the substances reported herein. Towards this goal, acute toxicity of organotin(II) and (IV)complexes is a drawback for further studies. In addition, from our previously reported results, in healthy cell lines (human lymphocytes cultures) [55], it became evident that **2a** was cytotoxic above the concentration of 20 μM, while its anti-PAF activity was in the micro-molar range (IC_50_ = 10.3 μM). Obviously, this organotin(II) complex holds the requirements for further in vivo investigation.

Finally, the results of the present study are in favor of the development of new metal-based agents with dual anti-PAF and anticancer activities in an era of increased interest.

## 3. Materials and Methods

### 3.1. General

Standard inert-gas atmosphere techniques were used for all syntheses and sample manipulations. All solvents were dried by standard methods, distilled under argon and stored over 4 Å molecular sieves (Å). All other chemicals were commercially available and used as received. Elemental analyses were obtained from the Central Analytical Group of the Chemistry Department of the University of Bonn. FT-IR spectra were recorded as KBr pellets and in ATR mode, in the region of 4000–400 cm^−1^, on a Shimadzu IR Affinity-1 spectrometer. All ^1^H, ^13^C{^1^H}, ^31^P{^1^H} and ^119^Sn{^1^H} NMR spectra were recorded on Bruker AM-300 and Bruker Avance Neo 400 MHz spectrometers in C_6_D_6_ or CDCl_3_. The ^1^H and ^13^C{^1^H} NMR spectra were calibrated against the internal residual proton and natural abundance ^13^C resonances of the deuterated solvent. The ^119^Sn{^1^H} NMR chemical shifts were referenced to Me_4_Sn. Each tin spectrum was acquired for about 0.5 h to 1 h. Melting points were determined using a Büchi 530 melting point apparatus and are not corrected. The samples were sealed under vacuum in capillary tubes and heated at a rate of 3 K min^−1^ to a temperature, which was lower by 10 K than the melting point.

#### Synthesis of the Organotin Complexes 2–3

*L_OEt_SnCl* (**2a**). Under an argon atmosphere, 843 mg (1.51 mmol) of NaL_OEt_ and 286 mg (1.51 mmol) of SnCl_2_ were mixed in a Schlenk tube. The tube was cooled at −60 °C and 40 mL of THF were added via a double-ended needle. The solution was allowed to warm at ambient temperature and stirred at ambient temperature over 5 h. The yellow suspension was evaporated to dryness and dried in vacuo for 1 h. The residue was dissolved in 15 mL of CH_2_Cl_2_, filtered and the clear yellow solution was concentrated to a few milliliters and stored at −60 °C for 1 h. The crystalline solid obtained was treated with 40 mL of cold pentane (−50 °C), leading after decantation to a light-yellow solid, which was dried in vacuo. Yield: 970 mg (93% relative to NaL_OEt_). m.p. 156 °C. Anal. Calcd for C_17_H_35_ClCoO_9_P_3_Sn (689.47 g.mol^−1^): C, 29.61; H, 5.12; Cl, 5.14. Found: C, 29.78; H, 5.24; Cl, 5.20%. FT-IR (KBr): ν[cm^−1^] = 3144(w) [ν(C=CH) of Cp], 3106(w), [ν(C=CH) of Cp], 2979(m), 2928(m), 2897(m), 2866(m), 1478(m), 1443(m), 1429(m), 1421(m), 1389(m), 1290(w), 1160(m), 1119(s) [*ν*(PO)], 1112 and 1094 (sh. vs), 1040 and 1013 (sh. vs), 941(vs), 844 (m), [*γ*(CH) of Cp], 775 and 751 and 721 (sh. s), 618(s), 591 and 576 (sh. vs), [δ(PO)], 496(m). ^1^H NMR (C_6_D_6_, 300 MHz, 298 K): *δ* 1.15 (t, ^3^*J*(H,H) = 7.1 Hz, 18H, C*H_3_*), 4.16 (m, 12H, C*H*_2_CH_3_), 4.88 (s, 15H, C_5_*H_5_*). ^31^P{^1^H} NMR (C_6_D_6_, 121.5 MHz, 298 K): *δ* 110.8. ^1^H NMR (CDCl_3_, 300 MHz, 298 K): *δ* 1.25 (t, ^3^*J*(H,H) = 7.1Hz, 18H, C*H_3_*), 4.12 (m, 12H, C*H*_2_CH_3_), 5.06 (m, 15H, C_5_*Me_5_*). ^31^P{^1^H} NMR (CDCl_3_, 121.5 MHz, 298 K): *δ* 111.6. ^13^C{^1^H} NMR (CDCl_3_, 75.5 MHz, 298 K): *δ* 10.0 (s, C_5_*Me_5_*), 16.5 (s, CH_3_), 61.6 (q, ^2^*J*(PC) + ^4^*J*(PC) = 6.2 Hz, *C*H_2_), 90.0 (s, *C_5_*H_5_). ^119^Sn{^1^H} NMR (C_6_D_6_, (111.9 MHz, 298 K):*−δ* = −314.0 (q, ^2^*J*^119/117^Sn,^31^P = 84–89 Hz).

*L*_OEt_SnCl* (**2b**). A solution of NaL*_OEt_ (417 mg, 0.66 mmol) in 10 mL of THF was added via a double-ended needle to a solution of SnCl_2_ (132 mg, 0.69 mmol) in 15 mL of THF cooled at −60 °C. The solution was allowed to warm at ambient temperature and stirred overnight. The yellow-orange suspension was evaporated to dryness and dried in vacuo for 2 h. The residue was dissolved in CH_2_Cl_2_ (15 mL), filtered and the clear yellow-orange solution was concentrated to a few milliliters giving an orange solution. Upon addition of cold (−50 °C) pentane (40 mL) to the orange solution cooled at the same temperature, a yellow-orange solid was precipitated that was allowed to settle for some minutes. After decantation, the yellow microcrystalline solid was dried in vacuo. Yield: 480 mg (95% relative to NaL*_OEt_). Recrystallization from 15 mL of diethyl ether at −78 °C, afforded 296 mg of complex **2b** analytically pure (59% yield). m.p. 199−199.5 °C. Anal. Calcd for C_22_H_45_ClCoO_9_P_3_Sn (759.61 g.mol^−1^): C, 34.79; H, 5.97; Cl, 4.67. Found: C, 34.63; H, 5.40; Cl, 4.88%. IR (KBr): ν [cm^−1^] = 2976(m), 2927(m), 2897(m), 2866(m), 1481(m), 1441(m), 1429(w), 1387(m), 1158(s), 1106(s) [ν(PO)], 1083 (sh.vs), 1030 and 1002 (sh. vs) 938(vs), 776 and 745 and 726 (sh. s), 625(s), 595 and 578 (sh. vs), [δ(PO)], 500(m). ^1^H NMR (C_6_D_6_, 300 MHz 298 K): *δ* 1.14 (t, ^3^*J*(H,H) = 7.2 Hz, 18H, C*H_3_*), 1.50 (s, 15H, C_5_*Me_5_*), 4.11 (m, 12H, C*H*_2_CH_3_). ^31^P{^1^H} NMR (C_6_D_6_, 121.5 MHz, 298 K): *δ* 111.2 ppm. ^13^C{^1^H} NMR (C_6_D_6_, 75.5 MHz, 298 K): *δ* = 10.3 (s, C_5_*Me_5_*), 16.4 (s, CH_3_), 61.3 (q, ^2^*J*(PC) + ^4^*J*(PC) = 5.9 Hz, *C*H_2_), 101.0 (s, *C_5_*Me_5_). ^1^H NMR (CDCl_3_, 300 MHz, 298 K): *δ* 1.16 (t, ^3^*J*(H,H) = 7.1Hz, 6H, 18H, C*H_3_*), 1.57 (m, 15H, C_5_*Me_5_*), 4.00 (m, 12H, C*H*_2_CH_3_). ^31^P{^1^H} NMR (CDCl_3_, 121.5, 298 K): *δ* = 112.6 ppm. ^13^C{^1^H} NMR(CDCl_3_, 75.5 MHz, 298 K): *δ* 10.0 (s, C_5_*Me_5_*), 16.0 (m, ^3^*J*(PC) = 1.8 Hz, *C*H_3_), 61.0 (q, ^2^*J*(PC) + ^4^*J*(PC) = 6.5 Hz, *C*H_2_ ), 101.2 (s, *C_5_*Me_5_). ^119^Sn{^1^H} NMR (C_6_D_6_, 111.9 MHz, 298 K): *δ* −302.6 ppm (q, ^2^*J*^119^Sn,^31^P = 103–105 Hz).

*L_OEt_SnPh_3_* (**3a**). A Schlenk tube, charged with an equimolar amount of NaL_OEt_ (460 mg, 0.82 mmol) and Ph_3_SnCl (320 mg) was cooled at −30 °C. Next, 25 mL of THF was added, the solution was allowed to warm at ambient temperature and then stirred overnight. During this time the clear yellow-orange solution became cloudy, and a white precipitate was formed. The suspension was then evaporated to dryness and dried in vacuo over 2 h. The residue was dissolved in 5 mL of CH_2_Cl_2_, filtered and the clear yellow-orange solution was concentrated to a few milliliters. Upon standing at ambient temperature a yellow solid started to crystallize, and the Schlenk tube was stored at −50 °C. Then 10 mL of cold (−50 °C) pentane was then added, and after filtration complex **3a** was obtained as a yellow solid. Yield: 550 mg (75% relative to NaL_OEt_). m.p. 231−232 °C. Anal. Calcd for C_35_H_50_CoO_9_P_3_Sn (885.33 g.mol^−1^): C, 47.48; H, 5.69. Found: C, 47.51; H, 5.49%. IR (KBr): ν [cm^−1^] = 3060(w) [ν(C=CH) of Cp], 3043(w) [ν(C=CH) of Cp], 2979(m), 2927(m), 2901(m), 1477(m), 1426(m), 1387(m), 1162(m), 1121(vs) [ν(PO)], 1088(sh.s), 1070(m), 1038 and 1026 (sh.s), 932(vs), 772(s), 729(s), 701(s), 653(m), 615 and 581 (sh. vs) [δ(PO)], 498(m), 457(m). ^1^H NMR (C_6_D_6_, 300 MHz, 298 K, ppm): *δ* 1.06 (t, ^3^*J*(H,H) = 7.3 Hz, 18H, C*H_3_*), 3.91 (m, 12H, C*H*_2_CH_3_), 4.88 (s, 5H, C_5_*H_5_*), 7.22–7.38 (m, 9H, *m* and *p*-Ph), 8.16 (d, ^3^*J*(^119^Sn,^1^H) = 69 Hz, ^3^*J*(^117^Sn,^1^H) = 55 Hz, 6H, *o*-Ph). ^31^P{^1^H} NMR (C_6_D_6_, 121.5 MHz, 298 K): *δ* = 111.5 (s). ^1^H NMR (CDCl_3_, 300 MHz, 298 K, ppm): *δ* = 1.09 (t, ^3^*J*(H,H) = 7.0 Hz, 18H, C*H_3_*), 3.78 (m, 6H, C*H*_2_CH_3_), 3.91 (m, 6H, C*H*_2_CH_3_), 5.00 (s, 5H, C_5_*H_5_*), 7.06–7.20 (m, 9H, *m* and *p*), 7.71 (d, ^3^*J*(^119^Sn,^1^H) = 69 Hz, ^3^*J*(^117^Sn,^1^H) = 57 Hz, 6H, *o*-Ph). ^31^P{^1^H} NMR (CDCl_3_, 121.5 MHz, 298 K): *δ* = 111.6 (s). ^13^C{^1^H} NMR (CDCl_3,_ 75.5 MHz, 298 K, ppm): *δ* 16.4 (q, ^3^*J*(^31^P,^13^C) + ^5^*J*(^31^P,^13^C) = 4.3 Hz, *C*H_3_), 60.8 (q, ^2^*J*(^31^P,^13^C) + ^4^*J*(^31^P,^13^C) = 6.1 Hz, *C*H_2_), 89.3 (s, *C_5_*H_5_), 126.2 (s, ^4^*J* (^119^Sn,^13^C) = 13.6 Hz, *p*-Ph), 126.5 (s, ^3^*J* (^119^Sn,^13^C) = 66 Hz, ^3^*J* (^117^Sn,^13^C) = 63 Hz, *m*-Ph), 136.8 (s, ^2^*J* (^119^Sn,^13^C) = 50 Hz, ^2^*J* (^117^Sn,^13^C) = 48 Hz, *o*-Ph), 155.3 (q, ^3^*J*(^31^P,^13^C) + ^4^*J*(^31^P,^13^C) = 7.4 Hz, *ipso*-Ph). ^119^Sn{^1^H} NMR (C_6_D_6_, 111.9 MHz, 298 K): *δ* = −401.4 (q, ^2^*J*(^119^Sn,^31^P = 84 Hz).

*L*_OEt_SnPh_3_* (**3b**). In a Schlenk tube, equimolar amounts of NaL*_OEt_ (299 mg, 0.48 mmol) and Ph_3_SnCl (183 mg) were added, and the solids were dried over 0.5 h in vacuo. The mixture was cooled at −30 °C and 25 mL of THF was added via a double-ended needle and the solution was allowed to warm to ambient temperature. During this time the clear yellow-orange solution became cloudy, and the suspension was stirred for 5 h. It was then evaporated to dryness and dried in vacuo over 2 h. The residue was dissolved in 5 mL of CH_2_Cl_2_, filtered and the clear yellow-orange solution was concentrated to a few milliliters. Upon standing at ambient temperature an orange solid started to crystallize, and the Schlenk tube was stored at −50 °C. Then 10 mL of cold (−50 °C) pentane was added, and after filtration complex **3b** was obtained as a yellow solid. Yield: 550 mg (75% relative to NaL*_OEt_). m.p. 277–278 °C. Anal. Calcd for C_40_H_60_CoO_9_P_3_Sn (955.46 g.mol^−1^): C, 50.28; H, 6.33. Found: C, 50.27; H, 6.07%. IR (KBr): ν [cm^−1^] = 3058(w), 3043(w), 2977(m), 1478(m), 1426(m), 1385(m), 1154(m), 1112(vs)[*ν*(PO)], 1098(s), 1069(s), 1042 (vs), 1018 (sh.s), 934(vs), 837(m), 775(m), 729(s), 701(s), 622 and 587 (sh. vs), [δ(PO)], 500(m). ^1^H NMR (C_6_D_6_, 300 MHz, 298 K, ppm): *δ* 1.05 (t, ^3^*J*(H,H) = 7.4 Hz, 18H, C*H_3_*), 1.55 (s, 15H, C_5_*Me_5_*), 3.88 (m, 12H, C*H*_2_CH_3_), 7.22–7.39 (m, 9H, *m* and *p*), 8.17 (d, ^3^*J*(^119^Sn, ^1^H) = 67.9 Hz, ^3^*J*(^117^Sn,^1^H) = 53.8 Hz, 6H). ^31^P{^1^H} NMR (C_6_D_6_, 121.5 MHz, 298 K): *δ* = 112.3 ppm. ^1^H NMR (CDCl_3_, 300 MHz, 298 K): *δ* = 1.08 (t, ^3^*J*(H,H) = 7.05 Hz, 18H, C*H_3_*), 1.64 (s,15H, C_5_*Me_5_*), 3.76 (m, 6H, C*H*_2_CH_3_), 3.87 (m, 6H, C*H*_2_CH_3_), 7.07–7.20 (m, 9H, *m* and *p*-Ph), 7.71 (d, ^3^*J*(^119^Sn,^1^H) = 68.1 Hz, ^3^*J*(^117^Sn,^1^H) = 52.3 Hz, 6H, *o*-Ph). ^31^P{^1^H} NMR (CDCl_3_, 121.5 MHz, 298 K): *δ* = 112.7 ppm. ^13^C{^1^H} NMR (CDCl_3_, 75.5 MHz, 298 K): *δ* 10.1 (s, C_5_*Me_5_*), 16.2 (d, ^3^*J*(^31^P,^13^C) = 1.8 Hz, *C*H_3_), 60.4 (q, ^2^*J*(^31^P,^13^C) + ^4^*J*(^31^P,^13^C) = 6.4 Hz, *C*H_2_), 125.8 (s, *p*-C), 126.2 (s, ^3^*J*(^119^Sn,^13^C) = 67.1 Hz, *m*-C), 136.9 (s, ^2^*J*(^119^Sn,^13^C) = 48.1 Hz, *o*-C), 155.3 (q, ^3^*J*(^31^P,^13^C) + ^4^*J*(^31^P,^13^C) = 6.9 Hz, ^1^*J*(^119^Sn,^13^C) = 745.9 Hz, ^1^*J* (^117^Sn,^13^C) = 719.3 Hz, *ipso*-Ph). ^119^Sn{^1^H} NMR (C_6_D_6_, 111.9 MHz, 298 K): *δ* −420.9 (q, ^2^*J*(^119^Sn,^31^P) *=* 84 Hz).

### 3.2. Single-Crystal X-ray Structural Determination

The X-ray structure analyses were performed on a STOE-STADI4 four-circle diffractometer and on a STOE-IPDS diffractometer using graphite monochromated Mo-Kα radiation (λ = 0.71073 Å). ψ-scan was carried out for (extinction coefficient = 0.0016(3)). Structure solution was performed with Direct Methods (SHELXS-97, SHELXS-86) and subsequent Fourier-difference synthesis (SHELXL-93, SHELXL-97) without restraints. Refinement on F^2^ was carried out by full-matrix least squares techniques. Non-hydrogen atoms were refined anisotropically. The hydrogen atoms were refined in **3b** isotropically (U_iso_ = 0.08) and in free. Hydrogen atoms were included using the riding model on the bound carbon atoms. A summary of the crystal data, data collection and refinement for the structures of the complexes **2a**, **2b** and **3b** is given in Table 2. CCDC 186,770 (**2b**), 186,771 (**3b**) and 186,772 (**2a**) contain the supplementary crystallographic data for this paper. These data can be obtained free of charge from the Cambridge Crystallographic Data Centre via www.ccdc.cam.ac.uk/data_reqeust/cif (accessed on 19 January 2023).

### 3.3. Biological Evaluation

#### 3.3.1. Materials and Methods for Biological Experiments in Washed Rabbit Platelets (WRPs)

The inhibitory effects against the thrombotic and inflammatory mediators, platelet activating factor (PAF) and thrombin in platelets for both tin(II) and tin(IV) coordination compounds, ligands and metal precursors were assessed against platelet aggregation of washed rabbit platelets (WRPs) and rabbit plasma rich in platelets (rPRP), as previously described [25,26,27]. The experiments were performed in a model 400 vs. aggregometer of Chrono-Log (Havertown, PA, USA) coupled to a Chrono-Log recorder at 37 ^°^C with constant stirring at 1200 rpm. The separation of platelets was performed with centrifugations in a 3L-R Heraeus Labofuge (Hvanau, Germany), a 400R Heraeus Labofuge and a RC-5B Sorvall (Newtown, CT, USA) refrigerated super speed centrifuge. Briefly, aliquots of standard PAF solution (Sigma, St. Louis, MO, USA) in chloroform/methanol (1:1, *v*/*v*) in glass tubes were evaporated under a stream of nitrogen and re-dissolved in Bovine Serum Albumin (BSA) (2.5 mg BSA/mL saline; Sigma, St. Louis, MO, USA) to obtain several concentrations (from 2.6 × 10^−8^ to 2.6 × 10^−11^ mol/L) of PAF solutions prior testing. Standard active thrombin (Sigma, St. Louis, MO, USA) was dissolved in saline prior to testing. The ligand precursors and metal complexes were dissolved in DMSO; therefore, the effect of DMSO was also assayed to WRPs. The ability of each selected sample to cause inhibition of either PAF-induced or thrombin-induced platelet aggregation was studied by adding various concentrations of each sample into the platelet suspension, just before the experiments. Details of the experimental protocol are described in the Appendix A (entry 19).

In cross-desensitization experiments against PAF-induced platelet aggregation, platelets were activated by a concentration of **2b** of its EC_50_ value, which is much higher (at least one to two orders of magnitude) than its IC_50_ value. After the complete disaggregation of the initial **2b**-induced platelet activation, platelets were stimulated by the addition of a PAF standard that can induce the same height aggregation curve. Vice versa, platelets were initially activated by an addition of a PAF standard in platelets, which induced a similar platelet aggegatory effect (similar height of aggregation curve) with that observed in the test by complex **2b**. These platelets were reactivated by a second stimulation, immediately after complete disaggregation of the initial PAF-induced platelet activation. For all the cross-desensitization tests synthetic PAF standard (16:0) in 4.4 ×10^−11^ M final concentration was used in the aggregometer cuvette. [53,54].

The PAF-like pathway-related aggregatory effects of samples on rabbit platelets are expressed as the concentrations of PAF that can induce the same effect of 50% of platelet aggregation, also known as the concentration effective in producing 50% of the maximal response (EC_50_ values) [63,64,65]. Biological assays were performed several times (*n* > 6) to ensure the validity of results. All experiments were also followed by appropriate control tests of the medium used (BSA, DMSO aliquots in BSA or saline), in both WRPs and rPRPs.

The platelets used for this study were from rabbits (about 20) of the research team of Emeritus Prof. of Biochemistry, C.A. Demopoulos. Ethical approval was received in accordance with the guidelines of the European Union for experimental animals.

#### 3.3.2. Cell Viability Assay

The Jurkat Cells (the culture was kindly donated by Prof. D. Galaris, Laboratory of Biological Chemistry, Faculty of Medicine, University of Ioannina, Greece), a permanent cell line that derived from a T lymphoma, were grown in suspension in RPMI 1640 growth medium, supplemented with 10% fetal serum (Gibco, Billings, MT, USA) and 100 units of penicillin/streptomycin (Pen-strep, Gibco). Cultures were kept in a water jacketed CO_2_ incubator (Forma Scientific, Marietta, OH, USA) at 37 °C. The tested compounds were dissolved in DMSO and were added to the cultures at the final concentrations of 1, 5, 10, 20, 50 and 75 μM. Duplicate cultures of 3 × 10^5^ cells were seeded in 24 well plates (Corning Incorporated, Corning, NY, USA). A cell count was performed per culture 17 h post-culture initiation, a period in which the cells are growing actively, and the various compounds at the designated concentrations were added in the culture medium. The cultures, under the influence of the tested compounds, were transferred in the incubator for three hours before counting the cells again. At 20 h post-culture initiation the growth medium was replaced by a fresh one, there was a recovery period and then the cells were again transferred into the incubator. The recovery process was designed to avoid false positive results during microscopy. At the end of 48 h post-culture initiation, a final cell count was performed to evaluate the percentage of viable cells. An initial control experiment using the known clastogenic factor Mitomycin (Mit-C, Sigma) at a final concentration of 1.5 μM was used to control the culture conditions [66]. Cisplatin, a known anti-proliferative compound, was used as positive control. The protocol applied is in accordance with Directive 487 (OECD 2010) for the cultivation of permanent cell lines.

## 4. Conclusions

A series of four organotin(II) and organotin(IV) complexes incorporating the Kläui oxygen tripodal ligand were designed, synthesized and biologically investigated against the PAF- and thrombin-induced aggregation in washed rabbit platelets. Additionally, cell viability assays in Jurkat cells were performed. The antiplatelet effects of the lead complexes 2b and 3a were significantly more potent than that of 1a and 1b, while 2b and 3a displayed a pronounced inhibitory activity against thrombin-induced aggregation. Complexes 2b and 3a also showed significant cytotoxicity effects in vitro on human Jurkat T lymphoblastic tumor cell line, higher than that of cisplatin. Consequently, the two organotin complexes could be potentially evaluated as candidates for new antitumor agents. Considering that 2b and 3a exhibit dual antiplatelet (anti-PAF) and cytotoxic activity, the in vivo administration of these complexes could be beneficial for cancer treatment, since it is well known from the literature that the administration of PAF inhibitors together with anticancer drugs improves the drug activity of the anticancer drug.

## Data Availability

The data presented in this study are available on request from corresponding author.

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
