# Peer review of "Tin(II) and Tin(IV) Complexes Incorporating the Oxygen Tripodal Ligands [(η5-C5R5)Co{P(OEt)2O}3]−, (R = H, Me; Et = -C2H5) as Potent Inflammatory Mediator Inhibitors: Cytotoxic Properties and Biological Activities against the Platelet-Activating Factor (PAF) and Thrombin"

_molecules, 2023, doi:10.3390/molecules28041859_

Round 1
Reviewer 1 Report
1. The title does not highlight the highlights of the article well. 2. Review all graphics, subtitles are small, ariel, no pattern. This seems irrelevant but it organizes the work for the reader.3. The word “ complex or compound” should be reunited in the text.
4. The IR data not discussed in the manuscript clearly. Some related refs could be highlighted, Inorganics, 10(2022) 20 and Micropor. Mesopor. Mat, 341(2022) 112098.
5. This methods part of the Cell lines and cytotoxicity assays should be illustrated clearly.
6. Please discuss the structure feature in detail, how about their weak interactions and packing interactions?
7. Why not compare the similar complexes on the biological activities against the Platelet-Activating Facto
Author Response
Responses to the referees’ comments.
Reviewer 1
- The title does not highlight the highlights of the article well.
Answer
Thank you for the kind suggestion. The title could be changed to “Tin(II) and tin(IV) complexes incorporating the oxygen tripodal ligand [(h5-C5R5)Co{P(OEt)2O}3]-, (R = H, Me; Et = -C2H5) as potent inflammatory mediator inhibitors: Cytotoxic properties and biological activities against the Platelet-Activating Factor (PAF) and Thrombin”.
- Review all graphics, subtitles are small, ariel, no pattern. This seems irrelevant but it organizes the work for the reader.
This has been done in the revised version. Thank you.
- The word “ complex “or compound” should be reunited in the text.
Answer
Both terms were used within the text according to the relevant part of the discussion. Complex, to emphasize on the organometallic chemistry part of the work and compound or substance for the biological assays. We thought that it was more correct like that. In the revised version we have changed it, as you have suggested.
- The IR data not discussed in the manuscript clearly. Some related refs could be highlighted, Inorganics, 10(2022) 20 and Micropor. Mesopor. Mat, 341(2022) 112098.
Answer
In the revised version, this part has been improved, according to your suggestion. According to your suggestion the suggested reference Micropor. Mesopor. Mat, 341(2022) 112098 has been included in the main text, that is reference [36a], of the revised manuscript.
- This methods part of the Cell lines and cytotoxicity assays should be illustrated clearly.
Answer
Thank you for the comment. However, we believe that this part is clearly demonstrated. It presents the experimental and discussion about the findings of this research. It has been performed according to well-reported methods from the laboratory of emeritus Prof. D. Matthopoulos, as reported in previous publications as well. Therefore, I do not see what could be changed further.
- Please discuss the structure feature in detail, how about their weak interactions and packing interactions?
Answer
Thank you for the comment. In the revised version, the structural features of all complexes were discussed further, including weak interactions. These are included in the supplementary Materials (Figures S3-S4 and S11).
- Why not compare the similar complexes on the biological activities against the Platelet-Activating Facto
Answer
This has been done within the text. Our discussion is separated into organotin(II) and organotin(IV) complexes, clearly shown that 2b and 3a are the most active, as presented in the conclusions part. Also you could see, for example line 311 of the revised version …”Since the final products 2b and 3a showed the lowest IC50 values against the PAF-induced aggregation of WRPs, it becomes evident that coordination of the bioactive oxygen tripodal ligand LOEt (LOEt*) into the metal precursors…” . However, to be in line with your suggestion we have added the following paragraph “Conclusively, we may propose that within this series the inhibitory activity drops in the following order: 2a > 3b > 3a 2b.” in Part 2.2.1.1
Reviewer 2 Report
Alexandros Kalampalidis et al. investigated on the synthesis and biological investigation of a series of organometallic tin(II) and tin(IV) complexes. The manuscript has interesting data and results for readers. The methods and results are adequately informed. However, before being considered further, sme points should be addressed.
1. The manuscript needs a detailed revision looking for some grammar, stylistic, and even typos issues.
2. The quality and resolution of figures and schemes must be improved.
3. The plots of NMR and FTIR spectra of the complexes 2a-b and 3a-b are missing In supplementary information.
4. The conclusion section summarizes the results. I recommend re-writing conclusions into conceptual findings from the mechanistic point of view.
Author Response
Responses to the referees’ comments.
Reviewer 2
Alexandros Kalampalidis et al. investigated on the synthesis and biological investigation of a series of organometallic tin(II) and tin(IV) complexes. The manuscript has interesting data and results for readers. The methods and results are adequately informed. However, before being considered further, sme points should be addressed.
- The manuscript needs a detailed revision looking for some grammar, stylistic, and even typos issues.
Answer
We thank the reviewer for his kind comment. In the revised version the manuscript has been thoroughly checked for some grammar, stylistic, and even typos issues.
- The quality and resolution of figures and schemes must be improved.
Answer
Schemes and Figures have been improved along with resolution.
- The plots of NMR and FTIR spectra of the complexes 2a-b and 3a-b are missing In supplementary information.
Answer
Thank you for the comment. Please see the message below I which has been sent to the Assistant Editor.
Comment to the Assistant Editor
I would like to be honest with you. I have finished corrections of the paper somedays ago; however, I realize a problem and I would like to share it with you. That is the following. According to the suggestion of a reviewer, the IR and NMR spectra of all complexes should be included in the Supplementary materials of the Revised version. As written in the supplementary materials, this work was performed some years ago, in the laboratory of my former supervisor in Bonn and are in a folder where all my spectra (original spectra) were included. However due to reconstructions in his laboratory, it is not possible to get these spectra within a short period of time. I have contacted him within the last days, explaining the problem, however I got the answer described before. For that reason, therefore I was unable to submit the revised version earlier. As a result, in the revised Supplementary materials, I have included FT-IR spectra of all complexes and 1H and 31P NMR spectra of 3a and 3b. Evidence about the existence of these complexes, if there are doubts about that, is undoubtably provided by the relevant NMR spectra of complexes 2a,2b,3a, 3b in DMSO-d6 shown in the Supplementary materials and from my laboratory book (relevant pages are available), where preparations of these complexes are described.
To this end, FT-IR spectra of 2a-3b and 1H, 31P NMR spectra of 3a and 3b, that were available, were added in the supplementary data (Figures S1-S2, S5-S6 and S7-S10).
- The conclusion section summarizes the results. I recommend re-writing conclusions into conceptual findings from the mechanistic point of view
Answer
Thank you for the comment. In the revised version a new paragraph has been added highlighting the inter-connection between PAF and cancer. This is the following “Considering that 2b and 3a exhibit dual antiplatelet (anti-PAF) and cytotoxic activity, the in vivo administration of these complexes could be beneficial for cancer treatment, since it is well known from the literature that the administration of PAF inhibitors together with anticancer drugs improves the drug activity of the anticancer drug”.
Reviewer 3 Report
The article generally presents a good chemical description of the synthesis of the indicated coordination compounds. In addition, the results of biological activity are encouraging, since it has been observed that the activity of some compounds obtained (3a and 2b) is higher than that of cisplatin.
The spectroscopic characterization of the synthesized compounds is correct. Although the 13C spectra are not shown in SI, they are commented on in the experimental part.
The reference J. Org. Chem. 2021, 86, 1311-1329, should be included in reference 1.
Author Response
Responses to the referees’ comments.
Reviewer 3
The article generally presents a good chemical description of the synthesis of the indicated coordination compounds. In addition, the results of biological activity are encouraging, since it has been observed that the activity of some compounds obtained (3a and 2b) is higher than that of cisplatin.
The spectroscopic characterization of the synthesized compounds is correct. Although the 13C spectra are not shown in SI, they are commented on in the experimental part.
Answer
- Thank you for the comment. Please see a general comment on that, which has been addressed to the Editor.
Comment to the Assistant Editor
I would like to be honest with you. I have finished corrections of the paper somedays ago; however, I realize a problem and I would like to share it with you. That is the following. According to the suggestion of a reviewer, the IR and NMR spectra of all complexes should be included in the Supplementary materials of the Revised version. As written in the supplementary materials, this work was performed some years ago, in the laboratory of my former supervisor in Bonn and are in a folder where all my spectra were included. However due to reconstructions in his laboratory, it is not possible to get these spectra (original spectra) within a short period of time. I have contacted him within the last days, explaining the problem, however I got the answer described before. For that reason, therefore I was unable to submit the revised version earlier. As a result, in the revised Supplementary materials, I could include only selective spectra of some compounds, not all of them. Evidence about the existence of these complexes, if there are doubts about that, is undoubtably provided by the relevant NMR spectra of complexes 2a,2b,3a, 3b in DMSO-d6 shown in the Supplementary materials and from my laboratory book (relevant pages are available), where preparations of these complexes are described.
To this end, FT-IR spectra of 2a-3b and 1H, 31P NMR spectra of 3a and 3b, that were available, were added in the supplementary data (Figures S1-S2, S5-S6 and S7-S10).
- The reference J. Org. Chem. 2021, 86, 1311-1329, should be included in reference 1.
Answer
According to your suggestion this reference has been included in reference 1.
Reviewer 4 Report
The manuscript „Tin(II) and tin(IV) complexes incorporating the oxygen tripodal ligand [(h5 -C5R5)Co{P(OEt)2O}3]− , (R = H, Me; Et = -C2H5): Cytotoxic properties and biological activities against the Platelet-Activating Factor (PAF) and Thrombin“ by Alexandros Kalampalidis and co-workers reports the synthesis and biological studies of a series of organometallic tin(II) and tin(IV) complexes with Kläui ligands. The manuscript is interesting but needs minor corrections. Specific points are listed below:
1. Line 50: A small error; the name organotin(IV) was split into two parts: "or" and "ganotin(IV)".
2. Line 332-336: Table 1 shows that ligand 1a had lower IC50 towards thrombin in WRPs than 2a and 2b complexes.
3. Line 341: Rather, 3b shows a 60-fold decrease in activity compared to 1b.
4. Line 443: The authors write: „More precisely, 2b exhibits a strong activity (lethal) at concentrations > 5 μM, while cell line viability (%) is significantly higher to that of cisplatin.” Rather, cell line viability (%) when using 2b (at concentrations > 5 μM) is lower than for cisplatin.
Author Response
Responses to the referees’ comments.
Reviewer 4
The manuscript „Tin(II) and tin(IV) complexes incorporating the oxygen tripodal ligand [(h5 -C5R5)Co{P(OEt)2O}3]− , (R = H, Me; Et = -C2H5): Cytotoxic properties and biological activities against the Platelet-Activating Factor (PAF) and Thrombin“ by Alexandros Kalampalidis and co-workers reports the synthesis and biological studies of a series of organometallic tin(II) and tin(IV) complexes with Kläui ligands. The manuscript is interesting but needs minor corrections. Specific points are listed below:
- Line 50: A small error; the name organotin(IV) was split into two parts: "or" and "ganotin(IV)".
Answer
This is due to spacing withing Word. Has been changed. Thank you.
- Line 332-336: Table 1 shows that ligand 1a had lower IC50towards thrombin in WRPs than 2a and 2b complexes.
Answer
We thank the reviewer for his valuable comment. We believe however, that the potency of 1a could only be compared to that of 2a and not 2b. Both complexes contain different ligand precursors, LOEt or LOEt*, around the tin(II) center, affording respectively, 2a and 2b. Thus, any comparison should be only made between 1a and 2a, 1b and 2b.
In fact, ongoing from 1a to 2a, the antithrombotic potency decreases by a factor of ten. Furthermore, we have realized that a similar trend is also followed in the case of the PAF-induced aggregation in WRPs (part 2.2.1.1). To our opinion this could be possibly attributed to the presence of the non-methylated ligand precursor LOEt, in the organotin(II) complex 2a, rendering this complex less potent against thrombin induced aggregation. This may also be the case for the six-co-ordinate complexes 3a, 3b when compare them with 1a, 1b. In other words, the biological activity drop in thrombin is almost the half for 3a compared to 1a and almost 70-fold from 3b to 1b, denoting that the presence of LOEt in 3a render it less potent.
Presumably the six-co-ordinate organotin complexes exert a higher affinity for the PAR receptors of thrombin, rendering them stronger inhibitors of the thrombin pathways as compared to 2a and 2b.
- Line 341: Rather, 3b shows a 60-fold decrease in activity compared to 1b.
Answer
See above, combined answer.
- Line 443: The authors write: „More precisely, 2b exhibits a strong activity (lethal) at concentrations > 5 μM, while cell line viability (%) is significantly higher to that of cisplatin.” Rather, cell line viability (%) when using 2b (at concentrations > 5 μM) is lower than for cisplatin.
Answer
Thank you for the kind suggestion. The sentence has been changed, in the revised manuscript, as suggested.
Round 2
Reviewer 1 Report
accepted.